# Prediction of Indirect Indicators of a Grass-Based Diet by Milk Fourier Transform Mid-Infrared Spectroscopy to Assess the Feeding Typologies of Dairy Farms

**DOI:** 10.3390/ani12192663

**Published:** 2022-10-04

**Authors:** Hélène Soyeurt, Cyprien Gerards, Charles Nickmilder, Jérôme Bindelle, Sébastien Franceschini, Frédéric Dehareng, Didier Veselko, Carlo Bertozzi, Nicolas Gengler, Antonino Marvuglia, Alper Bayram, Anthony Tedde

**Affiliations:** 1TERRA Research and Teaching Centre, Gembloux Agro-Bio Tech, University of Liège, 5030 Gembloux, Belgium; 2Walloon Agricultural Research Centre, Valorisation of Agricultural Products, 5030 Gembloux, Belgium; 3Comité Du Lait, 4651 Battice, Belgium; 4Walloon Breeders Association, 5590 Ciney, Belgium; 5Luxembourg Institute of Science and Technology (LIST), 41, Rue du Brill, L-4422 Belvaux, Luxembourg; 6Computational Sciences, Faculty of Science, Technology and Medicine, University of Luxembourg, 4365 Esch-sur-Alzette, Luxembourg; 7National Funds for Scientific Research, 1000 Brussels, Belgium

**Keywords:** milk, mid-infrared, grass, composition, grazing, spectrum, spectrometry

## Abstract

**Simple Summary:**

The dairy industry is interested in developing a detection tool for milk produced from pasture to certify the protected designation of origin of certain dairy products. As the cattle’s grazing influences the milk composition, the milk mid-infrared (MIR) spectra will be modified. Therefore, this study aims to develop a predictive model allowing one to use milk MIR spectrometry to classify milk as produced from a cattle diet either based on grass or not. To achieve this objective, the collection of grazing calendars from farms is needed. Unfortunately, this is hardly possible, as this information is rarely recorded. Therefore, the innovation of this research consists of using the usual farming practices developed in the southern part of Belgium combined with a large-scale milk database containing 48 milk composition-related traits. Indeed, in this geographical area, the cows are mainly on pasture between April and September. Therefore, the month of testing can be considered indirect information about their feeding. The developed models were able to distinguish with a precision of around 90% the supposed grass-based diet. Therefore, the probability of belonging to the GRASS class could be used in a tool counting the number of grazing days to confirm the labeling of dairy products.

**Abstract:**

This research aims to develop a predictive model to discriminate milk produced from a cattle diet either based on grass or not using milk mid-infrared spectrometry and the month of testing (an indirect indicator of the feeding ration). The dataset contained 3,377,715 spectra collected between 2011 and 2021 from 2449 farms and 3 grazing traits defined following the month of testing. Records from 30% of the randomly selected farms were kept in the calibration set, and the remaining records were used to validate the models. Around 90% of the records were correctly discriminated. This accuracy is very good, as some records could be erroneously assigned. The probability of belonging to the GRASS modality allowed confirmation of the model’s ability to detect the transition period even if the model was not trained on this data. Indeed, the probability increased from the spring to the summer and then decreased. The discrimination was mainly explained by the changes in the milk fat, mineral, and protein compositions. A hierarchical clustering from the averaged probability per farm and year highlighted 12 groups illustrating different management practices. The probability of belonging to the GRASS class could be used in a tool counting the number of grazing days.

## 1. Introduction

In Belgium, mainly in its southern part (i.e., the Walloon Region), as in other geographical areas such as New Zealand or Ireland, grassland represents a large part of the useful agricultural surface. In 2019, permanent grasslands represented 42% of the Walloon’s surface, and temporary grasslands represented 5% of the Walloon’s useful agricultural surface [1].

Grassland has an important place in our landscape and fulfills many roles. Grasslands have been shown to play an important role as a natural barrier against desertification through its root structure and soil cover [2]. Controlling its biomass through grazing allows for fire prevention and management by limiting the amount of fuel present [3]. Grassland is also a relatively stable carbon sink, as concluded in [4]. They calculated an average net annual productivity for the biome of 161 g of carbon per year per m^2^ over a 5-year study. In addition to these roles, grassland provides access to a lower-cost feed resource which requires minimal supplementation for dairy cows [5] and brings many important nutrients such as fiber and protein [6], which are essential for milk production.

Therefore, producing milk from pasture makes sense, and dairy companies have realized this. Indeed, many specifications for milk and products with protected designations of origin (PDOs) require a specified minimum grazing period for cows. For example, in France, the milk producer CANDIA recommends an average of 150 days of grazing per year for at least 6 h per day. The “Lait de pâturage” label is based on the same rules but also requires a minimum grazing area of 10 acres per animal. Finally, the “Grand Pâturage” milk requires access to a pasture for at least 180 days per year. In Belgium, the “MARGUERITE HAPPY COW” chain advocates feeding cows with at least 70% grass per ration and at least 180 days of grazing for a stocking rate of 4 animals per ha. The specifications for PDO products are sometimes a little vaguer. Indeed, for the PDO cheese “Abondance”, it is required that the herd’s ration be composed of at least 50% grass grazed during the summer and mostly hay during the winter, while the PDO cheese “comté” prohibits zero full grazing, and farmers must put their cows out to pasture as soon as possible and for as long as the weather permits. Finally, the specifications of the PDO cheese “Laguiole” specify having at least 120 days of grazing for the cows subscribing to this appellation. The common denominator of all those labels is the strong presence of a grass-based diet in the feeding rations of cows during a certain period of the year.

However, even though these specifications exist, there is no factual verification of these conditions. Thus, recently, the Chronopâture tool was developed to automatically count the number of days the herd spends grazing using a GPS collar. This tool has been tested on three farms with three different specifications and three different grazing times to be respected: milk grazing (120 days for 6 h), the PDO camembert (180 days), and the PDO “Beurre et Crème d’Isigny” (210 days) [7]. Unfortunately, such equipment can hardly be developed on a large scale, as it depends on the breeder’s choice. An alternative could be to observe more specifically the variations in milk composition according to the feed. Indeed, following [8], grass is the food richest in C18:3 fatty acids (FA). Therefore, its ingestion by cows will confer a particular composition to the milk. However, this composition is evolutionary. For example, the authors of [9] showed that the composition of the milk changed during the transition to pasture, with a progressive decrease in the quantity of milk produced (38%) and an increase in the fat (5%) and protein (5%) contents. In addition to the variation in the overall fat content of the milk, its composition is also impacted by grazing. Indeed, the milk of cows fed with fresh grass in pastures has a higher ratio of unsaturated to saturated fatty acids. Namely, it has a higher proportion of polyunsaturated fatty acids and conjugated fatty acids (CLA) than the milk of cows fed with a total mixed ration [10]. A fourfold increase in C18:3 polyunsaturated FAs in milk was observed when the milk was produced by grazing cows [8], while the concentration of conjugated linoleic acids (CLA) on pasture was twice as high as when the cows were fed with a total mixed ration [9]. Finally, there was also an increase in the long-chain FAs and a decrease in the short- and medium-chain FAs when the cows’ feeding was based on fresh grass [11]. Consequently, as FA contents can be predicted by milk Fourier transform mid-infrared (FT-MIR) spectroscopy, there is no doubt that we can observe changes in the milk FT-MIR spectra related to the presence of grass in the diet. For example, the authors of [12] demonstrated that it was possible to predict the cow’s ration composition and identify PDO practices using FT-MIR analysis of milk.

Unfortunately, collecting the grazing calendars of farms at a large scale is difficult because they are rarely recorded by the farmers. Therefore, the innovative approach of this work will consist of confirming if it is feasible to realize an indirect prediction of the presence of grass in the cattle diet by predicting the grazing period defined from the test month from a large database containing milk FT-MIR spectra. Then, the second question in this manuscript will consist of knowing if it is feasible to develop a farm’s large-scale feeding typology by using the estimated probability of cows grazing.

## 2. Materials and Methods

The software R version 4.1 (R Core Team, 2021) was used to compute all calculations. The data flow is summarized in Figure 1.

### 2.1. Data

The data came from the FuturoSpectre agreement linking the University of Liège—Gembloux Agro-Bio Tech (Gembloux, Belgium), the Walloon Research Centre (Gembloux, Belgium), the milk laboratory “Comité du Lait” (Battice, Belgium), and the Walloon Breeding Association (Ciney, Belgium). Data were collected from 2011 and 2021 at 2449 Walloon dairy farms as part of the milk payment scheme and represent 3,377,715 observations. The milk FT-MIR spectra were generated by analysis of the herd’s bulk milk using MilkoScan spectrometers (Foss Electric A/S, Hillerød, Denmark) at the milk laboratory “Comité du Lait” (Battice, Belgium) and standardized using the method developed in [13].

To improve the understanding of the prediction of grass-based diets from the test month, we decided to use 4 predictors given directly by the spectrometer (fat (%FAT), protein (%PROT), urea, and lactose contents) and 44 predictors derived from equations developed in the past by the team. Table 1 gives the specifications of those equations. FAs were formerly predicted in g/dL of milk and then turned into g/100 g of fat to capture the milk fat composition, which can be more informative in assessing the changes related to a grass-based diet.

### 2.2. Models

The dataset was split into calibration and validation sets. Given the large amount of available data, only 30% of farms were randomly included in the calibration set, keeping the others in the validation set. The model ought to predict the presence of grass in the diet. However, as no grazing calendars or feeding schemes were available, the innovative aspect of this work consisted of assessing the grass level indirectly by creating an indirect relationship with the trait based on the test month. In Wallonia, nearly all cows graze from April to September. However, grazing is an evolutive process, and the full grazing time globally takes place between May and August. Therefore, three different variables were created using the test month. To balance the modalities of those traits, the same number of months was used for the considered GRASS and NOGRASS periods. Those variables were named GRASS [i], with “i” being a placeholder taking value in the (1, 2, 3) set. GRASS1 considered the cows from May, June, July, and August as grass-based for feeding, while those in November, December, January, and February were not grass-based. All other month-related records were flagged as “others”. GRASS2 moved the May and November records into the “others” group, and GRASS3 further restricted the grass-based and not grass-based groups by also moving into “others” the June and December records. In conclusion, GRASS3 was a special case of GRASS2, which was a special case of GRASS1 (i.e., GRASS3 ⊂ GRASS2 ⊂ GRASS1).

According to prior conducted studies [17,18,19,20], the use of discriminant analysis using partial least squares (PLS DA) provides good predictive performance to detect the diets of cows. Therefore, through the R caret package [21], PLS DA was used to predict the created traits related to the period in which cows received a grass-based diet (i.e., GRASS, with NOGRASS and OTHERS removed). The optimal number of PLS DA latent variables was set based on a random cross-validation stratified following the 2 modalities using 10 groups. However, this number was bounded to not exceed 30 to limit the risk of over-fitting. The cross-validated sensitivity, specificity, and area under the curve (AUC) in receiver operating characteristic (ROC) analysis were estimated to assess the quality and robustness of the developed models. To validate the models on the same basis for all traits, the validation accuracy was calculated using only the records collected in December or January (i.e., NOGRASS) and in July or August (i.e., GRASS).

To assess the similarities between the three built models, Pearson correlation values were calculated between the probabilities of belonging to the GRASS modality obtained from the models. Finally, to interpret the reasons for predicting the GRASS modalities, the Variables Importance in Projection (VIP) score was calculated for each predictor and model. This score was based on the obtained PLS DA coefficients.

We can hypothesize that the probability value of belonging to a GRASS cluster is indirectly related to the grass growth, which is affected by the meteorological conditions. Therefore, significant correlations were expected between the GRASS probability and traits related to the meteorological conditions, and the humidity (%), the temperature (°C), the rate of clouds in the sky (%), as well as the amount of rain (mm) observed in Brussels were recorded for the entire studied period from the “Historique-Météo.net” website. The temperature humidity index was then calculated as follows: (0.8 × temperature) + ((humidity/100) × (temperature −14.4)) + 46.4. Finally, the Pearson correlations were calculated with those meteorological traits and the averaged GRASS probability estimated per year and month.

### 2.3. Feeding Typology of Farms

The working hypothesis is that the probability of belonging to the GRASS modality indirectly represents the level of grass in the diet. Therefore, this probability was studied throughout the year and also between farms. The probabilities were thus averaged per month and year to study the feeding typologies of the different farms. Hierarchical clustering using the Ward.D2 method was performed on the dataset. Then, after fixing the number of clusters from the heights observed in the dendrogram, the means of probability of the GRASS modality per month were averaged and plotted by cluster to highlight the differences between farms.

## 3. Results and Discussion

For all the built models, the number of PLS latent variables was equal to 30. Based on the AUC, sensitivity, and specificity values mentioned in Table 2, we can conclude that all models had a good ability to discriminate between the GRASS and NOGRASS modalities, which was confirmed by the external validation conducted, providing accuracies ranging from 89.35% to 90.54% (Table 2). 

As mentioned in [12,18], PLS DA has a very good ability to predict a cow’s diet from the milk FT-MIR spectrum. Very similar results (even if slightly lower) were obtained by replacing the FA content in milk with their related milk fat content (data not shown). The feeding fingerprint explains this capacity to discriminate the grass-based diet in milk FT-MIR spectral data. According to [8], fresh grass is the richest food in C18:3 FA. Therefore, its ingestion by cows confers a particular composition for the milk. A literature review [10] revealed that the milk of cows fed with fresh green forage has a higher ratio of unsaturated to saturated fatty acids and contains a higher proportion of polyunsaturated FAs and CLA than the milk of cows fed with a total mixed ration (TMR). Even if the prediction accuracy for those FA traits by FT-MIR spectroscopy is moderate (Table 1), the same trends were observed from our predicted FA values during the grazing period (Figure 2).

Moreover, an increase in long-chain FAs and a decrease in short- and medium-chain FAs in the milk fat when the feeding of cows was based on fresh grass was observed in [11]. Again, this was also confirmed by our predicted FA values (Figure 3).

The most informative traits were all related to the fat composition. Indeed, by decreasing order of importance, the most informative markers were the predicted contents of C18:2 cis-9, cis-12, C16:0, C10:0, C6:0, and medium-chain FA in the milk (Figure 4). The fat content, which is the fraction of milk that varies the most during the passage to pasture [22], was also an important trait, even if it was not the most informative. The presence of short- and medium-chain FAs in the most informative predictors was expected, as those FAs are produced in the mammary gland during the de novo synthesis. The long-chain FAs are instead derived from the ingested feed [17]. The lower importance of polyunsaturated and long-chain FAs depicted in Figure 4 was therefore related to the strong opposite relationship existing between these groups of FAs. For example, C16:0, the second-most-informative trait, had a higher content in pasture-based milk. Moreover, in line with [23], which highlighted the higher content in grazing cows’ milk of C17:0 and total trans C18:1, Figure 4 reveals the importance of these variables in detecting a grass-based diet (VIP score greater than 60).

In addition, we included in this study as predictors in the model other traits related to the milk mineral and protein composition. In particular, we used as grass diet detection indicators (1) the content of Ca, which proved to be informative in discriminating the GRASS and NOGRASS modalities, and (2) the contents of Mg and P, although with lower informative relevancy (Figure 3). The authors of [24] also observed the impact of grass-based diets on the contents of Ca and P in milk. However, these authors found no link with Mg. The casein composition and the protein content were also important for discriminating milk samples produced from grass-based diets (Figure 4). This was also highlighted in [24]. Even if we used traits predicted by FT-MIR spectrometry with a fluctuating accuracy in this study, the relevancy of the predicted traits to discriminating the GRASS and NOGRASS modalities agreed with the literature. The initial model used 48 predictors that could be reduced to 22 by applying a VIP threshold of 50 to keep the most informative variables. For all GRASS traits, the prediction accuracy decreased. For instance, for GRASS1, the sensitivity decreased from 87 to 78%, and the specificity decreased from 89 to 79%, which was significantly lower and encouraged keeping all 48 initial predictors.

Even if the discrimination between grass- and non-grass-based diets seems feasible, there is no sharp discontinuity between those feeding strategies in the yearly grazing cattle routine. Instead, the cows are gradually put to (or removed from) pasture, favoring the animals’ adaptation. Therefore, dichotomously affecting a record for either the “GRASS” or “NOGRASS” cluster would lack the transition-in and out-grazing information. The PLS DA can provide a probability of belonging to a specific cluster, allowing observation of the feeding transition period within and between farms. For all built models, we observed an increasing probability value for grass-based feeding from spring to summer and then a decreasing value (Figure 5), suggesting the models’ ability to recognize the feeding transition period despite not being trained using such examples. If the cows are fed with a higher proportion of grass (i.e., a higher probability value), this means that the grass growth has reached a sufficient level to support this need. Therefore, we can hypothesize that the probability value of belonging to the GRASS cluster is indirectly related to the grass growth. From Irish data, the authors of [25] observed that grass growth is highly affected by the temperature and solar radiation. Peak grass growth was observed in late spring and early summer. Then, the growth decreased in the late summer and autumn due to a decrease in temperature and solar radiation. This evolution is in line with the one observed from the probability values of belonging to the GRASS cluster (Figure 5). The meteorological conditions can therefore explain the variation in values observed between and within the years. The correlation between the GRASS probability and the relative humidity was low (−0.34). However, by combining the ambient temperature and the relative humidity into the temperature humidity index (THI), the obtained correlation was high (0.88) and equal to the one observed with the temperature, confirming the strongest influence of the heat on the probability values. As expected, a negative correlation (−0.56) was observed with the presence of clouds, which can influence grass growth. A small positive correlation was also calculated with the monthly amount of rain (0.15). Moreover, some proportion of the observed variability in the probability values could be related to changes in the floristic pasture composition and the grass composition itself.

The differences in the AUC values were small between models, suggesting a similarity between models’ performances (Table 2). Moreover, the correlation values, estimated from the entire validation set (N = 2,365,113), between the probabilities of belonging to the GRASS modalities were high between the 3 models (within the range [0.95, 0.98]), suggesting common information given by all models. However, by looking at the evolution of the probability throughout the studied years, we can observe that the trend for GRASS1 probability was slightly different from the two other trends (GRASS2 and GRASS3; Figure 4). This could be due to the more extensive range of months used to build GRASS1 model. Indeed, this could better cover the variability in grass quality and quantity. Unfortunately, no reference information is available to confirm this hypothesis. If this hypothesis is true, then the GRASS1 model must be more appropriate.

Using the developed models, we could estimate the probability of belonging to the GRASS cluster for each test day to distinguish the feeding patterns between farms. The obtained probabilities for GRASS1 were averaged by farm, year, and month. Only the farms having 12 values per year (i.e., full-year representation) were studied, and their averaged probability values for belonging to the GRASS class were introduced in hierarchical clustering. The obtained dendrogram is illustrated in Figure 6, and the respective descriptive statistics are mentioned in Table 3. The annual trend of their average probability values within the year is depicted in Figure 7. Twelve groups could be isolated, with some of them being very close. Indeed, clusters 1, 4, and 5 were related to the farms having a feeding ration for more intensive production (Figure 7).

Those clusters were related to higher milk production and a lower content of long-chain FAs. The farms belonging to this group of clusters could move within those clusters between years. Therefore, the differences between 1, 4, and 5 could mainly be related to external factors such as the meteorological conditions, leading to different grass production, or the economical context of the feed prices, which could also influence the kind of feeding. However, as expected, based on the annual trend depicted by those clusters, we can hypothesize that they also put their cows out to pasture during the grazing period. Indeed, in each pattern, we can observe an increasing probability value from spring to summer.

However, the intensity was different, which can probably be explained by the meteorological conditions influencing the grass production and the potential economical context, which can influence the feed purchase. Cluster 8 had a constant annual evolution (Figure 7) with a high probability value. However, based on Table 3, this group would be related to farms that have analyzed skimmed milk. Indeed, the level of fat was nearly equal to zero. By extracting the fat from the milk, the milk matrix was modified. Therefore, the predictions performed for those spectral data could not be considered. This cluster was therefore not interesting. Clusters 6, 7, 2, 11, and 10 would be related to a group of farms given a feed ration for more extensive production. The feed will be richer in long-chain FAs, as confirmed by the higher probability values observed for those farms during the non-grazing period. The last group of clusters was harder to classify, as cluster 12 corresponded to intensive farms (Table 3) whose production mainly occurred during 2021. This year had a different pattern than the others (Figure 5), with an annual evolution showing a rapid increase in grazing probability (Figure 7). This could be related to adequate meteorological conditions for grass growth and quality. However, this hypothesis must be confirmed by experts. Clusters 3 and 9, which were siblings of cluster 12, seemed to be an intermediate status between the previous groups (Table 3). Consequently, we could detect the presence of grazing by estimating the GRASS probability. The model could also assess the farm’s intensive or extensive practices throughout the allotted time. Suppose we set to −1 the “intensive” group (i.e., clusters 4, 5, and 1) and to 1 the “extensive” group (i.e., clusters 6, 7, 2, 11, and 10), and “intermediate” group (i.e., clusters 3, 12, and 9). In that case, we could sum those values per year to appreciate the intensive character of a farm. A total of 1601 farms had 11 years of data collection. Of those farms, only 6.8% were always in the intensive group, and 4.3% of them were in the extensive group. The remaining farms were intermediate. As mentioned in [26], some farms can also make a transition between intensive and extensive characteristics.

## 4. Conclusions

The development of an easy method of detection of milk produced from grass-based diets is of interest due to the labeling existing in the dairy market for promoting this kind of healthier milk product. The innovative aspect of this study consisted of using usual farming practices in the southern part of Belgium to define the traits related to grazing. The assumption of the grass-based diet from the month of milk testing seemed to be relevant after examining the information relevancy of each predictor with regard to the literature. The good discrimination of the GRASS milk samples was mainly related to the changes in the milk fat composition and the modifications of the mineral and protein contents. Even if the models were not trained using milk samples produced during the feeding transition periods, they were able to highlight the evolution to a full grass diet thanks to the increase in the probability value for belonging to the GRASS modality. By averaging this value by farm, month, and year, we confirmed the different feeding strategies existing in the southern part of Belgium, where nearly all herds realized grazing but with different levels of intensity. Even if the global annual trend were similar, the intensity can vary due to meteorological conditions. The composition of the pasture could also influence the probability value. However, to confirm this statement, the collection of field information would be necessary. This will be the objective of our future work. Having those results first will allow us to select the farms of interest more efficiently. Moreover, using predictors instead of milk MIR spectra allowed for a better interpretation of the occurring discrimination. This was important to have an indirect validation of the models’ relevancy. Finally, the long-term objective of the research described in this paper is the creation of a tool that will be able to automatically estimate the grazing periods of cattle down to counting the exact number of days spent by cows on pasture.

## Figures and Tables

**Figure 1 animals-12-02663-f001:**
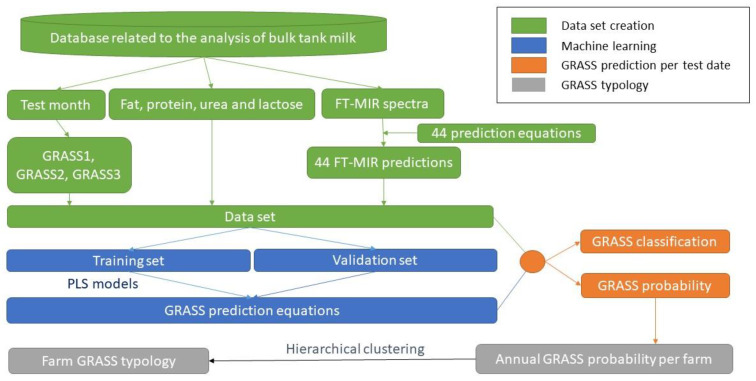
Data flow.

**Figure 2 animals-12-02663-f002:**
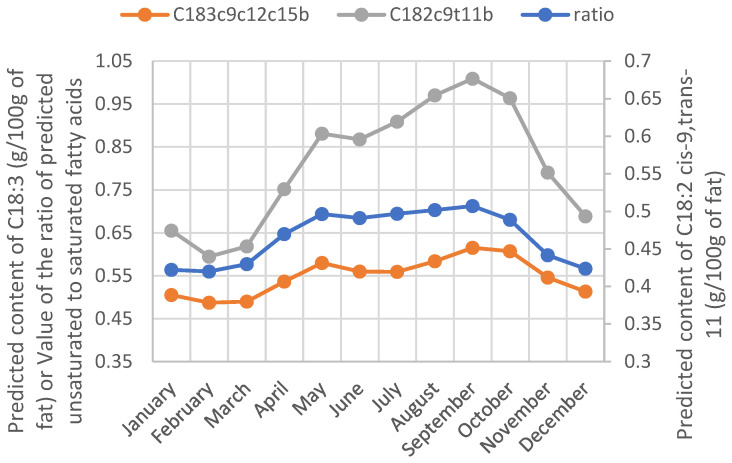
Yearly evolution of the ratio of predicted unsaturated to saturated fatty acids and the predicted content of C18:2 cis-9, trans-11 and C18:3 cis-9, cis-12, cis-15 in milk fat.

**Figure 3 animals-12-02663-f003:**
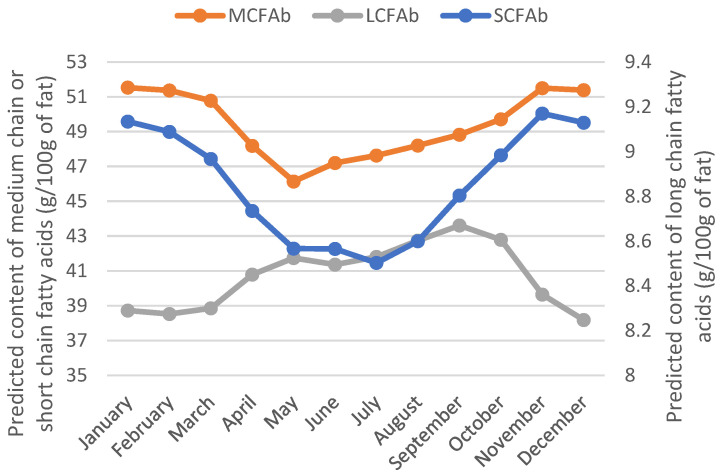
Yearly evolution of the predicted content of short-, medium-, and long-chain fatty acids in milk fat.

**Figure 4 animals-12-02663-f004:**
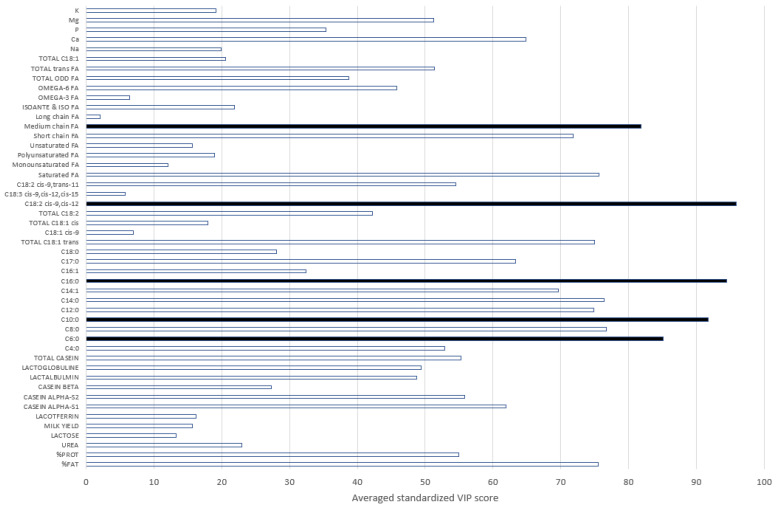
Scores of the Variable Importance in Projection (VIP) for all predictors and models. Black bars represent the most important traits (FA = fatty acids).

**Figure 5 animals-12-02663-f005:**
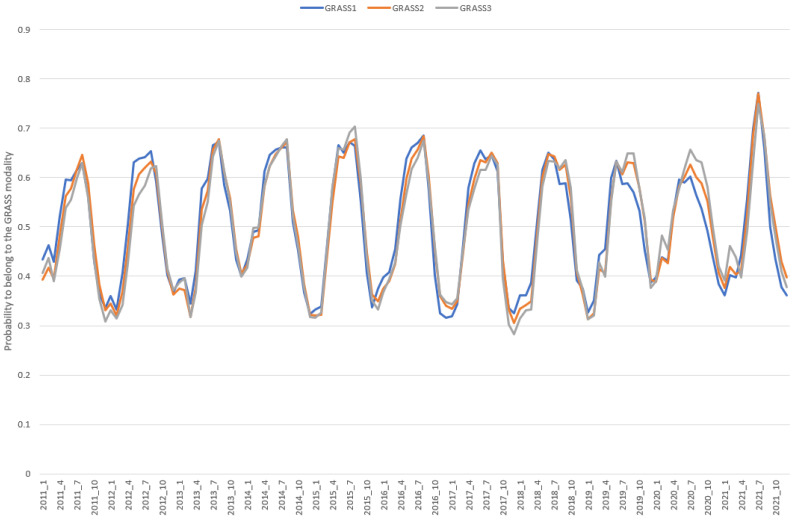
Evolution of the probability of belonging to the GRASS modality for the 11 studied years based on the validation set (N = 2,365,113 records).

**Figure 6 animals-12-02663-f006:**
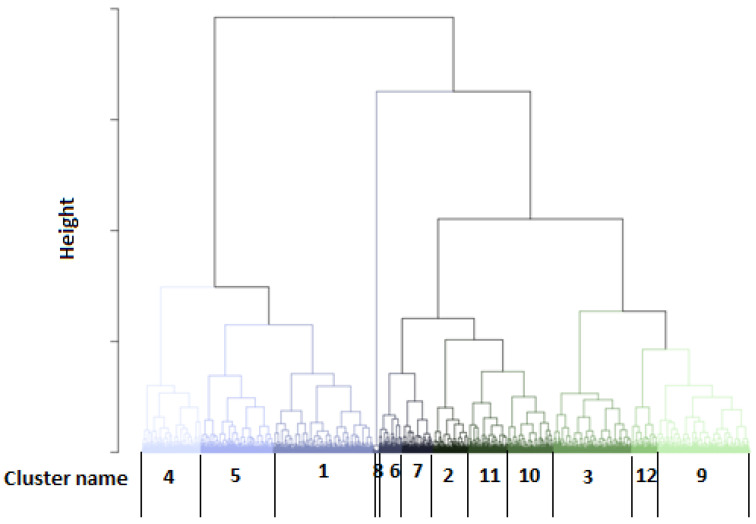
Dendrogram established from hierarchical clustering using the average monthly probability values for belonging to the GRASS1 cluster.

**Figure 7 animals-12-02663-f007:**
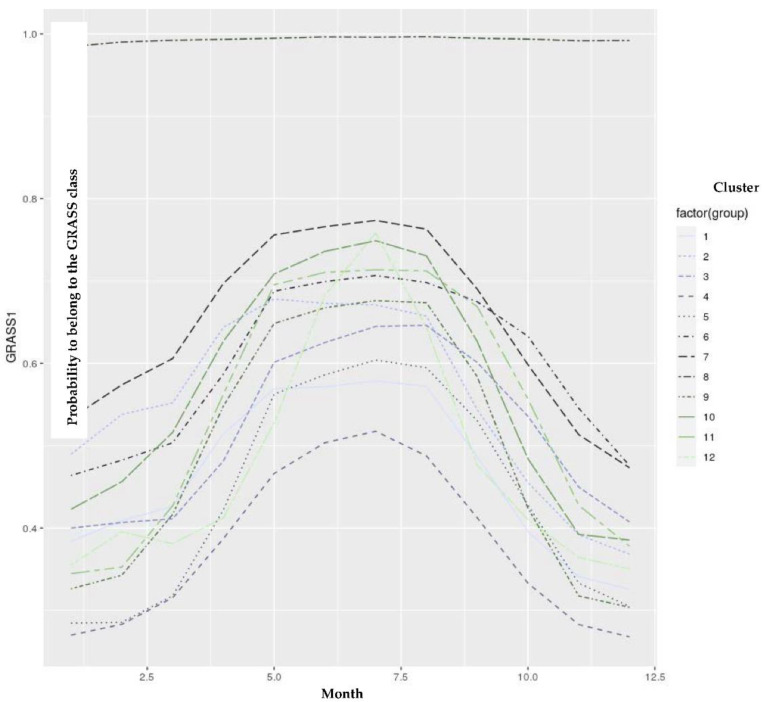
Dendrogram established from hierarchical clustering using the average monthly probability values for belonging to the GRASS1 cluster.

**Table 1 animals-12-02663-t001:** Specifications of equations used to provide the used predictors.

Traits	Unit	N ^1^	R^2^	RMSE ^2^	Ref. ^3^
Milk yield	kg/day	457	0.69	3.48	NP ^4^
C4:0	g/dL lait	1371	0.93	0.008	[14]
C6:0	g/dL lait	1371	0.91	0.006	[14]
C8:0	g/dL lait	1371	0.91	0.004	[14]
C10:0	g/dL lait	1371	0.92	0.010	[14]
C12:0	g/dL lait	1371	0.93	0.011	[14]
C14:0	g/dL lait	1371	0.94	0.030	[14]
C14:1 cis	g/dL lait	1371	0.71	0.008	[14]
C16:0	g/dL lait	1371	0.95	0.091	[14]
C16:1 cis	g/dL lait	1371	0.73	0.013	[14]
C17:0	g/dL lait	1371	0.81	0.003	[14]
C18:0	g/dL lait	1371	0.84	0.056	[14]
Total of C18:1	g/dL lait	1371	0.96	0.060	[14]
Total of C18:1 trans	g/dL lait	1371	0.80	0.025	[14]
Total of C18:1 cis	g/dL lait	1371	0.95	0.063	[14]
C18:1 cis-9	g/dL lait	1371	0.95	0.061	[14]
Total of C18:2	g/dL lait	1371	0.71	0.014	[14]
C18:2 cis-9, cis-12	g/dL lait	1371	0.75	0.011	[14]
C18:2 cis-9, trans-11	g/dL lait	1371	0.74	0.010	[14]
C18:3 cis-9, cis-12, cis-15	g/dL lait	1371	0.69	0.004	[14]
SFA	g/dL lait	1371	0.99	0.072	[14]
MUFA	g/dL lait	1371	0.97	0.059	[14]
PUFA	g/dL lait	1371	0.79	0.021	[14]
UFA	g/dL lait	1371	0.97	0.064	[14]
SCFA	g/dL lait	1371	0.93	0.025	[14]
MCFA	g/dL lait	1371	0.97	0.104	[14]
LCFA	g/dL lait	1371	0.95	0.110	[14]
Branched FA	g/dL lait	1371	0.77	0.013	[14]
Total of omega-3	g/dL lait	1371	0.68	0.006	[14]
Total of omega-6	g/dL lait	1371	0.74	0.014	[14]
Total of odd FA	g/dL lait	1371	0.84	0.016	[14]
Total of trans FA	g/dL lait	1371	0.82	0.029	[14]
Lactoferrin	mg/L milk	5541	0.55	139.01	[15]
Casein alpha-s1	g/L milk	135	0.81	0.58	NP
Casein alpha-s2 + K	g/L milk	135	0.81	0.36	NP
Casein beta	g/L milk	133	0.75	1.13	NP
Lactalbumin	g/L milk	138	0.38	0.15	NP
Lactoglobuline	g/L milk	134	0.81	0.25	NP
Total of casein	g/L milk	133	0.84	1.56	NP
Sodium (Na)	mg/kg of milk	1019	0.44	50.98	[16]
Calcium (Ca)	mg/kg of milk	1094	0.82	53.38	[16]
Phosphorus (P)	mg/kg of milk	1083	0.75	58.71	[16]
Potassium (K)	mg/kg of milk	1090	0.55	88.14	[16]
Magnesium (Mg)	mg/kg of milk	1124	0.72	6.53	[16]

^1^ N = number of records. ^2^ RMSE = root mean squared error. ^3^ Ref = reference. ^4^ NP = not published.

**Table 2 animals-12-02663-t002:** Performances of discriminant analysis performed on the three created traits related to the grazing season. The validation set contained 721,192 records.

	TenFold Stratified Cross-Validation				Validation
	N	AUC ^1^	Sensitivity	Specificity	Accuracy
GRASS1	533,786	94.71 ± 0.07	86.78 ± 0.17	88.61 ± 0.17	89.66
GRASS2	397,409	96.21 ± 0.08	88.41 ± 0.27	90.84 ± 0.19	90.95
GRASS3	265,876	97.43 ± 0.08	90.55 ± 0.18	92.99 ± 0.21	91.40

^1^ AUC = area under the curve.

**Table 3 animals-12-02663-t003:** Descriptive statistics for the farm clustering (SFA = saturated fatty acids, MUFA = monounsaturated fatty acids, and LCFA = long-chain fatty acids).

Cluster Name	N Sample	% Sample	Milk	% Fat	% Protein	g/100 g Fat		
			kg/Day	g/100 g	g/100 g	SFA	MUFA	LCFA
1	4239	16.41	26.46	4.14	3.46	69.14	26.73	39.55
2	1498	5.80	25.98	3.96	3.40	67.24	28.41	41.94
3	3316	12.84	26.20	4.09	3.43	68.22	27.72	40.68
4	2483	9.61	26.90	4.33	3.55	70.24	25.35	37.28
5	3092	11.97	26.37	4.24	3.49	69.26	26.52	39.18
6	1321	5.11	25.51	3.89	3.38	66.85	29.01	42.34
7	989	3.83	25.22	3.74	3.36	65.49	30.35	44.52
8	257	0.99	28.34	0.25	3.54	50.05	45.55	63.37
9	3902	15.11	25.49	4.12	3.44	67.91	27.54	40.87
10	1954	7.56	25.45	3.95	3.40	66.31	28.82	42.29
11	1660	6.43	25.41	4.04	3.41	67.27	28.55	42.01
12	1121	4.34	27.75	4.21	3.48	67.53	27.19	38.63

## Data Availability

The data presented in this study are available on request from the corresponding author.

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
