# Peer review of "Prediction of Indirect Indicators of a Grass-Based Diet by Milk Fourier Transform Mid-Infrared Spectroscopy to Assess the Feeding Typologies of Dairy Farms"

_animals, 2022, doi:10.3390/ani12192663_

Round 1

Reviewer 1 Report

The innovative aspect of the study is that it provided a way to make use of large amounts of DHI data that is accumulating in dairy labs and created a prediction model for a specific application, which is the grazing status of a herd, without the need for additional data related to the rations or the feeding regimes, which can be very challenging to collect and usually require full cooperation of farmers. Also, samples in the transition period were not needed for training the model and yet they were predicted correctly. Such specific samples are usually not easy to obtain to be included in the model.

Points to improve:

Create a diagram for the methodology or elaborate more in the Materials and Methods to better understand what was predicted from what. The article mentions milk spectra, 48 milk traits, test month and probability of being GRASS. However, it is difficult to keep up with the text to figure out which was predicted from what. I had to read the paper multiple times to figure out what I think was done in this study. Better explanation of the methodology will be highly appreciated.

Line 109 the innovative approach of this work will consist of realizing an indirect prediction of the level of grass in the cattle diet by predicting the test month from a large database containing milk FT-MIR spectra. Then, the probability of cows having been grazing will be used to reflect the level of grass in the diet and will be studied to assess the farms' large-scale feeding typology.

Comment: After reading the entire paper, I did not see where in the manuscript the test month was predicted from milk FT-MIR spectra. Please clarify whether milk spectra were used to predict the test month or not.

Line 125 To improve the understanding of the prediction of grass-based diets from the test month, we decided to use 4 predictors given directly by the spectrometer (fat (%FAT), protein (%PROT), urea, and lactose contents) and 44 predictors derived from equations developed in the past by the team.

Comment: The author did not clearly demonstrate how the grass-based diets were predicted from the test month. If the author refers to the prediction of the probability of being GRASS in a specific month, then they should clarify what exactly is being predicted from the models, which is in this case a probability of belonging to a class.

Line 138 The model ought to predict the level of grass in the diet.

Comment: When using the word level, it implies a percentage of grass or some other quantitative number to describe the portion of grass in the diet. However, in the body of the paper, the author only described the performance of qualitative models that predicted either the class of “grass” or “nograss”. If the author refers to the probability of being GRASS, as mentioned in line 172, then they should use that word instead of using the term “level of grass in the diet”.

Line 143 Therefore, 3 different variables were created using the test month.

Comment: This line contradicts with line 109, which states that test month was predicted from milk spectra. My understanding after reading the paper, that test month was used to assign a record to the GRASS class in the three classification criteria.

Line 158 The optimal number of PLS DA 158 latent variables was set based on a random stratified cross validation following the 2 modalities using 10 groups.

Comment: What were the strata based on? Was it based on herd or something else?

Line 156 Therefore, through the R caret package [21], PLS DA was used to predict the created traits related to the period in which cows received a grass-based diet 157 (i.e., GRASS and NOGRASS, OTHERS were removed). The optimal number of PLS DA latent variables was set based on a random stratified cross validation following the 2 modalities using 10 groups. However, this number was bounded not to exceed 30, to limit the risk of over-fitting.

Comment: It is not clear what the predictors were for the PLS-DA model. Were they milk spectra as mentioned in Line 109 or the 48 milk traits predicted from milk spectra as mentioned in line 125?

Line 176 Then, after fixing the number of clusters from the heights observed in the dendrogram, the means of probability of GRASS modality per month were averaged and plotted by cluster to highlight the difference between farms.

Comment: The author does not specify the criteria to choose the appropriate height to set the number of the clusters in the dendrogram accordingly. If such criteria were used, it should be mentioned.

Line 233 Even if in this study we used traits predicted by FT-MIR spectrometry having a fluctuating accuracy, the relevancy of predicted traits to discriminate GRASS and NOGRASS modalities agreed with the literature.

Comment: Here the author mentions that the prediction of GRASS and NOGRASS was done based on the 48 milk traits. It was only here that the author made it clear that the classification was based on the 48 milk traits.

Line 237 However, by looking at the evolution of the probability throughout the studied years, we can observe that the trend for GRASS1 probability was slightly different from the 2 other trends (GRASS2 and GRASS3; Figure 4).

Comment: The author claims that the probability of belonging to the GRASS class can be used later to predict the number of grazing days as stated in line 46 in the abstract “The probability of belonging to GRASS class could be used in a tool counting the number of grazing days.”. However, Figure 4 shows that the probabilities for the 3 classification traits, GRASS1, GRASS2 and GRASS3 are similar in general, despite the differences in the number of grassing days for each classification method, specially for peaks at July 2013, 2014, 2016 and 2021. Also, no differences in probabilities are noticed on the sides of the peaks, between the minima and the maxima between the 3 classification criteria. The author did not clarify how those probabilities can be used as a basis for the calculation of the number of grazing days, when the 3 different criteria for GRASS have different number of grazing days but producing very similar probabilities.

Line 259

Comment: the author correlates meteorological conditions with probabilities of being GRASS or NOGRASS. However, this part of this research was not described in the Materials and Method section. Also, the author did not clarify the relevance or the importance of such observed correlations and the purpose that they serve in the suggested tool to classify milk as GRASS or NOGRASS. The intensity of the available grass, which is a result of meteorological conditions, might affect how much grass a cow can get per day, but will it affect the number of grazing days and the probability of being GRASS? I ask this question because the intention of the author is to create a tool to count grazing days.

Line 288 Indeed, clusters 1, 4, and 5 are related to the farms having a feeding ration for a more intensive production (Figure 5).

Comment: How did the author verify this information? In general, the author did not mention enough information about how they performed the cluster analysis. It would be appreciated to give more details on how this analysis was performed.

Line 288 Indeed, clusters 1, 4, and 5 are related to the farms having a feeding ration for a more intensive production (Figure 5).

Line 289 Therefore, the differences between 1, 4, and 5 can be mainly related to external factors like the meteorological conditions leading to a different grass production or the economical context of the feed prices which could also influence the kind of feeding. However, as expected, based on the annual trend depicted by those clusters, we can hypothesize that they represented cows on pasture. Indeed, in each pattern, we can observe an increasing probability value from spring to summer.

Comment: In those two paragraphs, the author claims that clusters 1,4 and 5 have “feeding ration for a more intensive production” and “they represented cows on pasture”. Does that mean that herds in these 3 clusters were on intensive grazing system? It would be appreciated to clarify this point better. Also, there is a connection made to the meteorological conditions and economical context; however, no data were shown to support these claims and nothing was mentioned in the Materials and Methods on how they got reached these conclusions.

Line 352 Moreover, using predictors instead of milk MIR spectra allowed a better interpretation of the occurring discrimination.

Comment: This statement in conclusion contradicts line 110 “predicting the test month from a large database containing milk FT-MIR spectra”. After reading the manuscript, I think milk spectra were only used to predict the 48 milk traits that were later used in the PLS-DA prediction model. Therefore, it would be highly appreciated to clarify the section Methods and Materials and maybe have a diagram or any other means to clearly describe the methodology that was used in this study.

Author Response

Reviewer 1

AU: Dear reviewer, thank you for the time spent to read our manuscript and formulate such interesting comments to improve the manuscript.

The innovative aspect of the study is that it provided a way to make use of large amounts of DHI data that is accumulating in dairy labs and created a prediction model for a specific application, which is the grazing status of a herd, without the need for additional data related to the rations or the feeding regimes, which can be very challenging to collect and usually require full cooperation of farmers. Also, samples in the transition period were not needed for training the model and yet they were predicted correctly. Such specific samples are usually not easy to obtain to be included in the model.

Points to improve:

Create a diagram for the methodology or elaborate more in the Materials and Methods to better understand what was predicted from what. The article mentions milk spectra, 48 milk traits, test month and probability of being GRASS. However, it is difficult to keep up with the text to figure out which was predicted from what. I had to read the paper multiple times to figure out what I think was done in this study. Better explanation of the methodology will be highly appreciated.

AU: Thank you for this suggestion. A scheme reflecting the data flow is now added (Figure 1 and Line 117)

Line 109 the innovative approach of this work will consist of realizing an indirect prediction of the level of grass in the cattle diet by predicting the test month from a large database containing milk FT-MIR spectra. Then, the probability of cows having been grazing will be used to reflect the level of grass in the diet and will be studied to assess the farms' large-scale feeding typology.

Comment: After reading the entire paper, I did not see where in the manuscript the test month was predicted from milk FT-MIR spectra. Please clarify whether milk spectra were used to predict the test month or not.

AU: You are right the sentence was not enough clear. GRASS traits were defined using the test month. Therefore, it is an indirect prediction of the test month. The sentence was rewritten as follow for more clarity: “Therefore, the innovative approach of this work will consist of confirming if it is feasible to realize an indirect prediction of the presence of grass in the cattle diet by predicting the grazing period defined from the test month from a large database containing milk FT-MIR spectra.” (Lines 108-111).

Line 125 To improve the understanding of the prediction of grass-based diets from the test month, we decided to use 4 predictors given directly by the spectrometer (fat (%FAT), protein (%PROT), urea, and lactose contents) and 44 predictors derived from equations developed in the past by the team.

Comment: The author did not clearly demonstrate how the grass-based diets were predicted from the test month. If the author refers to the prediction of the probability of being GRASS in a specific month, then they should clarify what exactly is being predicted from the models, which is in this case a probability of belonging to a class.

AU: The data flow is now added and should improve the clarity. The definition of GRASS is explained at lines 150-158. The GRASS definition is only based on the test month. For instance, for GRASS1: records with a test month in May, June, July or August received a value of 1 meaning “grass-based diet”, records with a test month in November, December, January, February received a value of 0 for GRASS1 meaning “no grass-based diet”. The other records received the label “OTHER”. The objective of this study is to use the label 0 and 1 to develop an algorithm to discriminate grass-based diet and no grass-based diet. I hope that the scheme related to the data flow will clarify this aspect.

Line 138 The model ought to predict the level of grass in the diet.

Comment: When using the word level, it implies a percentage of grass or some other quantitative number to describe the portion of grass in the diet. However, in the body of the paper, the author only described the performance of qualitative models that predicted either the class of “grass” or “nograss”. If the author refers to the probability of being GRASS, as mentioned in line 172, then they should use that word instead of using the term “level of grass in the diet”.

AU: You are right. The term “level” was replaced by “presence” (Lines 143). We have also made this change in the introduction (Lines 108-111).

Line 143 Therefore, 3 different variables were created using the test month.

Comment: This line contradicts with line 109, which states that test month was predicted from milk spectra. My understanding after reading the paper, that test month was used to assign a record to the GRASS class in the three classification criteria.

AU : This line 109 was not enough clear. The past sentence was modified as mentioned in the previous comment and a scheme representing the data flow was added (Figure 1).

Line 158 The optimal number of PLS DA 158 latent variables was set based on a random stratified cross validation following the 2 modalities using 10 groups.

Comment: What were the strata based on? Was it based on herd or something else?

AU : You are right. It was not enough clear. The cross-validation was stratified according to the labels (grass or not grass). The sentence was rewritten as follows: “The optimal number of PLS DA latent variables was set based on a random cross valida-tion stratified following the 2 modalities using 10 groups.” (Lines 163-165)

Line 156 Therefore, through the R caret package [21], PLS DA was used to predict the created traits related to the period in which cows received a grass-based diet 157 (i.e., GRASS and NOGRASS, OTHERS were removed). The optimal number of PLS DA latent variables was set based on a random stratified cross validation following the 2 modalities using 10 groups. However, this number was bounded not to exceed 30, to limit the risk of over-fitting.

Comment: It is not clear what the predictors were for the PLS-DA model. Were they milk spectra as mentioned in Line 109 or the 48 milk traits predicted from milk spectra as mentioned in line 125?

AU : The predictors were the 48 milk predicted traits. The scheme reflecting the data flow should improve the clarity (Figure 1).

Line 176 Then, after fixing the number of clusters from the heights observed in the dendrogram, the means of probability of GRASS modality per month were averaged and plotted by cluster to highlight the difference between farms.

Comment: The author does not specify the criteria to choose the appropriate height to set the number of the clusters in the dendrogram accordingly. If such criteria were used, it should be mentioned.

AU : We have not used a specific criteria. The number of cluster was fixed by observing the dendrogram representing the heights. No optimization for the number of cluster was used.

Line 233 Even if in this study we used traits predicted by FT-MIR spectrometry having a fluctuating accuracy, the relevancy of predicted traits to discriminate GRASS and NOGRASS modalities agreed with the literature.

Comment: Here the author mentions that the prediction of GRASS and NOGRASS was done based on the 48 milk traits. It was only here that the author made it clear that the classification was based on the 48 milk traits.

AU : I hope that the scheme related to the data flow will improve the clarity of the applied methodology (Figure 1).

Line 237 However, by looking at the evolution of the probability throughout the studied years, we can observe that the trend for GRASS1 probability was slightly different from the 2 other trends (GRASS2 and GRASS3; Figure 4).

Comment: The author claims that the probability of belonging to the GRASS class can be used later to predict the number of grazing days as stated in line 46 in the abstract “The probability of belonging to GRASS class could be used in a tool counting the number of grazing days.”. However, Figure 4 shows that the probabilities for the 3 classification traits, GRASS1, GRASS2 and GRASS3 are similar in general, despite the differences in the number of grassing days for each classification method, specially for peaks at July 2013, 2014, 2016 and 2021. Also, no differences in probabilities are noticed on the sides of the peaks, between the minima and the maxima between the 3 classification criteria. The author did not clarify how those probabilities can be used as a basis for the calculation of the number of grazing days, when the 3 different criteria for GRASS have different number of grazing days but producing very similar probabilities.

AU : This is another research topic and it was not the scope of the present document. However to answer to your question, the possibility is to fixed a threshold of probability meaning a grass-based diet then you can make a count of test date with a probability higher to this threshold. However, depending of the season, this threshold could be different. Another possibility should be to take the increase of probability per farm. If the increase is sufficient, we could count a grazing period. However, to develop that, we must have grazing information (grazing calendars). This information was not available for the current study. We are trying now to collect such data but it is not easy.

Line 259

Comment: the author correlates meteorological conditions with probabilities of being GRASS or NOGRASS. However, this part of this research was not described in the Materials and Method section. Also, the author did not clarify the relevance or the importance of such observed correlations and the purpose that they serve in the suggested tool to classify milk as GRASS or NOGRASS. The intensity of the available grass, which is a result of meteorological conditions, might affect how much grass a cow can get per day, but will it affect the number of grazing days and the probability of being GRASS? I ask this question because the intention of the author is to create a tool to count grazing days.

AU : To answer your questions with a sufficient detail, we need to have grazing information. However, this was not available. You are right this information was not given in the material and methods. It is now added (Lines 176-185).

Line 288 Indeed, clusters 1, 4, and 5 are related to the farms having a feeding ration for a more intensive production (Figure 5).

Comment: How did the author verify this information? In general, the author did not mention enough information about how they performed the cluster analysis. It would be appreciated to give more details on how this analysis was performed.

AU : The scheme related to the data flow should improve the clarify (Figure 1). The intensification of studied farms was not verified as no data are available. It is a hypothesis.  

Line 288 Indeed, clusters 1, 4, and 5 are related to the farms having a feeding ration for a more intensive production (Figure 5).

Line 289 Therefore, the differences between 1, 4, and 5 can be mainly related to external factors like the meteorological conditions leading to a different grass production or the economical context of the feed prices which could also influence the kind of feeding. However, as expected, based on the annual trend depicted by those clusters, we can hypothesize that they represented cows on pasture. Indeed, in each pattern, we can observe an increasing probability value from spring to summer.

Comment: In those two paragraphs, the author claims that clusters 1,4 and 5 have “feeding ration for a more intensive production” and “they represented cows on pasture”. Does that mean that herds in these 3 clusters were on intensive grazing system? It would be appreciated to clarify this point better. Also, there is a connection made to the meteorological conditions and economical context; however, no data were shown to support these claims and nothing was mentioned in the Materials and Methods on how they got reached these conclusions.

AU : You are right. It was not clear. The farms belonging to those cluster are more intensive. However, the cows are on pasture also during the grazing period. This is very common in the Walloon Region of Belgium. The sentence was modified to bring more clarity: “However, as expected, based on the annual trend depicted by those clusters, we can hypothesize that they put also their cows on pasture during the grazing period.”. (Lines 326-328)

Line 352 Moreover, using predictors instead of milk MIR spectra allowed a better interpretation of the occurring discrimination.

Comment: This statement in conclusion contradicts line 110 “predicting the test month from a large database containing milk FT-MIR spectra”. After reading the manuscript, I think milk spectra were only used to predict the 48 milk traits that were later used in the PLS-DA prediction model. Therefore, it would be highly appreciated to clarify the section Methods and Materials and maybe have a diagram or any other means to clearly describe the methodology that was used in this study.

AU: This is not fully true as all predictors are coming from milk FT-MIR spectra. A scheme related to the data flow was added to improve the clarity (Figure 1).

Reviewer 2 Report

I like this work a lot and have only a few aspects that should be reconsidered:

1) "So, this study aims to develop...." ? Is this study finished or still running ? So, is only a pre-result or incremental parts of an overall result.

2) Due the separation "simple summary", "abstract", "introduction", all these more differentiation and sharpening. Often abstracts are placed away from the main text (e.g., in databases), this needs also storytelling. Introduction should end in amore clearer scientific hypothesis and far more clearer aims ? Of course it is mentioned slightly in the "simple abstract", but in the introduction it needs to be more comprehensively and scientifically.

3) Labelling of Figures 3 and 4 and 6 is very small. Labelling of Figure 5 is weird.

4) In some parts, it needs the help of a native speaker. For example, "In addition, we included in this study as predictors in the model other traits related  to the milk mineral and protein composition" ??? What do you want to tell ?

Author Response

I like this work a lot and have only a few aspects that should be reconsidered:

AU: Thank you for the interest given to this study and your comments.

  • "So, this study aims to develop...." ? Is this study finished or still running ? So, is only a pre-result or incremental parts of an overall result.

AU : The algorithm is finished but now we tried to find reference data to validate it. Therefore, the sentence was kept.

  • Due the separation "simple summary", "abstract", "introduction", all these more differentiation and sharpening. Often abstracts are placed away from the main text (e.g., in databases), this needs also storytelling. Introduction should end in amore clearer scientific hypothesis and far more clearer aims ? Of course it is mentioned slightly in the "simple abstract", but in the introduction it needs to be more comprehensively and scientifically.

AU : I’m not sure to understand what you want for the rewriting of the abstract and simple abstract. Could you provide me more details? The final paragraph of the introduction was rewritten to better formulate the scientific questions as you suggest (Lines 108-113).

  • Labelling of Figures 3 and 4 and 6 is very small. Labelling of Figure 5 is weird.

AU : This comment was not formulated by the other reviewers. Have you some concrete suggestions to improve the figures by keeping the same information?

  • In some parts, it needs the help of a native speaker. For example, "In addition, we included in this study as predictors in the model other traits related to the milk mineral and protein composition" ??? What do you want to tell ?

AU : As mentioned by the other reviewer, it seems to be difficult to follow the methodology in the first version of this article. Therefore, a scheme representing the data flow was added to have a better clarification. (Figure 1)